# African Swine Fever (ASF) Trend Analysis in Wild Boar in Poland (2014–2020)

**DOI:** 10.3390/ani12091170

**Published:** 2022-05-03

**Authors:** Maciej Piotr Frant, Anna Gal-Cisoń, Łukasz Bocian, Anna Ziętek-Barszcz, Krzysztof Niemczuk, Anna Szczotka-Bochniarz

**Affiliations:** 1Department of Swine Diseases, National Veterinary Research Institute, Partyzantów Avenue 57, 24-100 Puławy, Poland; anna.gal@piwet.pulawy.pl (A.G.-C.); anna.szczotka@piwet.pulawy.pl (A.S.-B.); 2Department of Epidemiology and Risk Assessment, National Veterinary Research Institute, Partyzantów Avenue 57, 24-100 Puławy, Poland; lukasz.bocian@piwet.pulawy.pl (Ł.B.); anna.barszcz@piwet.pulawy.pl (A.Z.-B.); 3Director General, National Veterinary Research Institute, Partyzantów Avenue 57, 24-100 Puławy, Poland; krzysztof.niemczuk@piwet.pulawy.pl

**Keywords:** ASF, wild boar, seropositive, endemic

## Abstract

**Simple Summary:**

African swine fever (ASF) has been present in Poland since 2014. The article describes and explains the changes in the ASF epidemic in the wild boar population in the period 2014–2020. In that relatively short time, the disease has spread to about half of the territory of Poland, affecting eastern and western provinces. Most ASF-positive animals were molecular/virus-positive, however, the observation of the increase of serologically positive animals (potential survivors) in successive years of the epidemic, especially in areas where the virus has been present for a longer time, may indicate the potential beginning of ASF endemicity in Poland.

**Abstract:**

African swine fever (ASF) is a lethal hemorrhagic disease of *Suidae*, i.e., domestic pigs and wild boars. The disease was introduced to Poland in 2014 and is now present in the wild boar population. Appropriate ASF prevention requires further research for answers to fundamental questions about the importance of vectors in virus transmission, the impact of environmental factors on the presence of ASFV in wild boar habitats, and the role of survivors as potential virus carriers and their part in the potential endemicity of ASF. In order to analyze the changes in the molecular and serological prevalence of ASFV in wild boar population in Poland, real-time PCR and ELISA/IPT tests were conducted. In the analyzed period (2014–2020), most of the ASF-positive wild boars were molecular/virus-positive, however, over the years the percentage and the number of seropositive animals has increased. At the beginning of the epidemic, the disease was limited to a small area of the country. Since then, it has spread to new provinces of Poland. From the beginning and until today, most notifications of ASF-positive wild boars were for carcasses (passive surveillance), however, the number of serologically positive animals is still increasing. Despite the fact that notifications of ASF outbreaks are still being received near the eastern border of Poland, the old ASF area seems to be limited mainly to ASF serologically positive animals, which may indicate the beginning of ASF endemicity in Poland.

## 1. Introduction

African swine fever (ASF) is a highly contagious, hemorrhagic disease caused by a dsDNA virus belonging to the *Asfivirus* genus and is the only known member of *Asfaviridae* family [1,2]. Twenty-four genotypes of African swine fever virus (ASFV) have been identified. The division is mainly dependent on differences in the major capsid protein (p72) encoded by the B646L gene [2,3]. All European ASFV strains are classified to genotype II. This type of the virus is highly virulent. In the affected European regions, an acute form of disease is recorded, with 100% mortality among domestic pigs and wild boar [4].

ASFV affects domestic pigs (*Sus scrofa domestica*), Eurasian wild boar (*Sus scrofa*), and the African wild *Suidae* family. Infected animals show a wide range of symptoms depending on virus isolate, the route and dose of infection, and the immunological status of the animal. Clinical signs are very similar in wild boar and pigs [1,5]. The disease can take different forms depending on the virulence of the strain. Peracute and acute forms are observed when infection is caused by highly virulent strains, while moderately virulent strains induce acute and subacute forms. The chronic form of the disease develops with infection by moderate-to-low virulence strains [5,6].

ASF has affected many regions of the world for over 100 years [7]. It was first reported in Kenya in 1921. By the 1950s, the disease had been noted in eastern and southern Africa and the southern parts of Central Africa. In 1959, an outbreak appeared in West Africa [8]. In 1957, ASF appeared in Lisbon. After 1960, the virus spread to the whole Iberian Peninsula, where it has been present for more than 30 years [8]. In this time, outbreaks affected other European countries such as Belgium, France, Malta, Italy, and the Netherlands [9].

The second wave of the disease started in 2007 when ASF was detected in Georgia. Genotype II of ASFV has spread rapidly to neighboring countries, the Russian Federation, Ukraine, and Belarus. In the first half of 2014, it reached parts of the European Union (EU)—Lithuania, Poland, Latvia, and Estonia [10]. The following years brought new cases of the disease in Czechia, Hungary, Romania, Moldova, Bulgaria, Belgium, Slovakia, Greece, Serbia, and Germany [1,7]. Since 2018, the disease has expanded to China, which is the world’s largest pig producer [11]. In recent years, the wave of the epidemic has swept through further Asian countries [1].

In July 2021, Estonia reported its first ASF outbreak in domestic pigs since 2017. Between February 2019 and August 2020, only ASFV-seropositive wild boar were detected. After this time, positive wild boar began to appear in the central part of Estonia [12]. A similar decrease in the number of infected wild boar was observed in Latvia. In Lithuania, despite efforts, ASF is still spreading in the wild boar population and outbreaks on farms are reported every year [10]. In Poland in 2021, a large number of outbreaks in domestic pigs were reported in eleven provinces out of sixteen (Figure 1). ASF is still present in the wild boar population [13].

According to Gervasi et al. (2021), the epidemic proceeds in four successive phases, where the first is the invasion phase of an ASF-free area. This phase is the point in time with the highest chance for elimination of ASFV. After 1–2 years from the initial invasion, the first wave of the epidemic takes place, lasting 3–4 years, with a rapid spread of the virus. Thereafter, the epidemic switches to an endemic phase (5–7 years). The fourth phase is the second wave of the epidemic, which occurs within 8–10 years [1].

There are four well-known epidemiological ASF cycles, appearing in different geographical regions. Transmission happens by specific vectors in all cycles. The oldest, sylvatic cycle, encountered in African countries, runs with the participation of wild reservoirs, such as warthogs and soft ticks of *Ornithodoros* spp. [3,14]. ASF is detected in all development stages of ticks and is transmitted both sexually and transtadially [2]. The second transmission cycle, tick–>pig, occurs without the presence of other members of the *Suidae* family. The third cycle is domestic, where the virus is transmitted among domestic pigs that became infected due to feeding with contaminated swill. The final one is the wild boar–habitat cycle, during which the virus is transmitted to wild boar by contaminated pig-derived products and carcasses present in natural wild boar habitats and maintains in the wild boar population [3,14,15]. In these last two methods of virus transmission, the tick vectors are not required [2].

Researchers continue to study the role of arthropods in ASFV transmission in Europe [16]. Observations of summer disease peaks in some areas may suggest the presence of a vector adequate to *Ornithodoros* spp. present in Africa and the Iberian Peninsula [1,7]. In Estonia, arthropods found in wild boar habitats were checked for ASFV (9% of the hunted wild boar in Estonia were positive in the analyzed period). No ASFV DNA was detected in any samples, with one fifth showing the presence of pig genetic material [17]. Other arthropods such as flies, mosquitoes, and some biting insects were observed and studied as potential ASF virus vectors. Although the viral DNA was found on the surface of the flies’ bodies, it was considered to be the result of the high contamination with ASFV of the surroundings of the tested flies. Therefore, it has been considered a minor risk for active participation in long-distance transmission of the virus [7,18]. So far, no evidence has been provided to support the hypothesis of a significant contribution of arthropods to the spread of ASFV epidemics in Eurasia [1,17]. In Russia and Eastern Europe, in a progressive epidemic of ASF, the participation of wild boar as a vector of disease transmission and virus reservoir is most important [15].

There are separate groups of animals at different stages of the course of infection: animals in the early phase of infection (3–10 dpi-days post-infection) with a high risk of virus transmission; animals with confirmed presence of ASFV and seropositive (7–10 dpi and up to 100 dpi), which carry some transmission risk and may still die of ASF; and the last group, with only seropositive animals which are potential survivors [18]. In this group, the virus is not detected in most of the samples, apart from some lymphatic tissues in a dose that does not constitute a risk of infection between survivors and susceptible animals [1,11,18]. In the EU, most animals still do not gain immunity in time and die, but healthy, seropositive wild boar are also noticed. In some African ecosystems, all samples from warthogs are only seropositive [2,7]. In Russia, Lithuania, and Estonia, ASFV isolates from wild boar are reported to cause infected pigs to develop a course of infection of up to 21 days, they suffer asymptomatically or can even completely fight off the infection [4]. It suggests that the circulation of ASFV within a wild boar population has contributed to the appearance of moderate- and low-virulent virus strains with reduced lethality [1,19]. This phenomenon is related to changes such as large deletions, duplications, and sequence transpositions between the genomes of the strains [20,21]. The host’s innate immune response, which is able to control virus replication and inhibit the development of disease, may also be important [2]. Survivor animals could be the main reason for which a disease remains persistent in some areas and may become endemic. It is associated with the periodic appearance of disease outbreaks and virus transmission to areas not affected by the disease. Due to the high survival rate of the virus in various environmental conditions even for several months, including tissues of infected animals, the risk of disease transmission is very high and persists over a long period of time [6].

Appropriate ASF prevention requires further research for answers to fundamental questions about the importance of vectors in virus transmission, the impact of environmental factors on the presence of ASFV in wild boar habitats and especially the role of survivors as potential virus carriers. It is also important to conduct further molecular epidemiology studies into the phylogeny and evolution of ASFV [18,22]. When the disease is present for a long time in the same area, the animals may adapt to virus pressure in the environment. The changes observed in the serological results during the analyzed period might prove potential endemicity of the ASF in Poland.

## 2. Materials and Methods

### 2.1. Sampling Methods

Whole laboratory analysis, from sample preparation, extraction of DNA to molecular and serological studies, was conducted in a biosafety 3 laboratory (BSL-3) in the National Reference Laboratory for ASF in Poland by qualified staff members (technicians, researchers).

The material used in the analysis were field (environmental) samples from wild boars collected in the years 2014–2020 within the framework of the ASF monitoring program in Poland (ASF zones 0, I, II, III). The material for the tests were blood/serum, bone marrow and tissue samples (e.g., spleen, kidney, tonsil, lung, lymph nodes). The samples were collected by the local Veterinary Inspection employees (as a part of the ASF monitoring program). All samples were analyzed for the presence of ASFV DNA and/or antibodies against ASFV using molecular and serological methods.

### 2.2. PCR Technique

The molecular detection of ASFV DNA was conducted with the use of the real-time PCR (qPCR) method. Before the analysis, sample preparation was performed. Tissues were homogenized in phosphate-buffered saline (PBS) yielding a 10% homogenate. Next, DNA extraction with the use of a QIAamp DNA Mini Kit (QIAGEN, Hilden, Germany) or QIAcube HT system (Indical, Leipzig, Germany) was performed. The positive control of DNA extraction was provided by the European Union Reference Laboratory (EURL) for ASF (CISA-INIA, Valdeolmos, Spain).

Real-time PCR was conducted by the method described by Fernandez-Pinero [23] or with the use of one of the commercial kits: Virotype (Indical, Leipzig, Germany), ID Gene African Swine Fever Duplex (IDvet, Grabels, France). The amplification process was conducted in one of five types of thermocyclers (Stratagene Mx 3005P Real-Time PCR System, Stratagene, La Jolla, CA, USA; Applied Biosystems 7500, Applied Biosystems, Waltham, MA, USA; QuantStudio™ 5, ThermoFisher Scientific, Waltham, MA, USA; Rotor-Gene Q, QIAGEN, Hilden, Germany; LightCycler 480, Roche, Basel, Switzerland). A fluorescent signal with a threshold cycle value (Ct) below 37.0 was considered as positive.

### 2.3. Serological Analysis

For serological analysis, serum was obtained from whole blood by centrifugation in 896–1514× *g*. The serological status of the serum samples was determined by enzyme-linked immunosorbent assay (ELISA). For that method, one of two commercial kits was used (ID Screen^®^ African Swine Fever Indirect, IDVet Innovative diagnostic, Grabels, France; Ingezim PPA COMPAC, Ingenasa, Madrid, Spain). The method was applied according to the manufacturer’s manual. 

All positive and doubtful results obtained in ELISA test were verified with the use of one of the confirmation tests: the indirect immunoperoxidase technique (IPT) or immunoblotting technique (IB). Test procedures (IPT and IB) are more sensitive and specific than ELISA. In the case of IPT, the result is observed in a reverse-field microscope. In the case of IB, the sera show a specific pattern of reaction in the nitrocellulose strips. Both the IPT and IB reagents were provided by the EURL for the ASF (CISA-INIA, Valdeolmos, Spain) and the analyses were conducted according to EURL protocols.

### 2.4. Surveillance

For the surveillance calculation, each molecular/virus-positive result (qPCR; ASFV DNA), serologically positive result (anti-ASFV antibody; ELISA/IPT), and molecular/virus-positive plus serologically positive (double positive results in qPCR and ELISA/IPT) were considered separately. For the further investigation, all obtained results were divided in six main groups of the wild boars from Poland (2014–2015):molecular/virus-positive wild boars (only qPCR analysis);molecular/virus-positive and serologically negative wild boars (qPCR and ELISA/IPT);molecular/virus-positive and serologically positive wild boars (qPCR and ELISA/IPT);molecular/virus-negative wild boars (only qPCR analysis);molecular/virus-negative and serologically negative wild boars (qPCR and ELISA/IPT);molecular/virus-negative and serologically positive wild boars (qPCR and ELISA/IPT).

All samples were analyzed for virus/DNA detection (qPCR). Only serum/blood samples could be analyzed by both methods: qPCR and ELISA/IPT.

Examination of the frequency of ASF in wild boars during the whole epidemic was estimated separately for each year.

Moreover, examination of the prevalence of ASF was conducted separately for wild boars found in different conditions (found-dead, road-killed, hunted) in different ASF zones (0, I, II, III) for all analyzed wild boars and for the wild boars analyzed in both (molecular/virus and serological) tests. Each wild boar whose carcass was found in the environment after the natural death of the animal was included in the found-dead group. In the road-killed group were wild boars killed in traffic accidents (killed by cars). The hunted group were animals shot by professional hunters from the Polish Hunting Association.

ASF zone 0 was the area where ASFV was not present and there were no legal restrictions connected to the ASF zoning. Restricted zone I was the area where ASFV was not present, however, a legal limitation connected with ASF was implemented there. Restricted zone II was the area where ASFV was present in the wild boar population. Restricted zone III was the area where ASF outbreaks in domestic pigs were confirmed. The ASF zones (0–I–II–III) were designed according to the 2014/709/EU decision [24], however on 21 April 2021 a new regulation for ASF zones was implemented. The new legislation which is currently in force is: 2021/605/EU [25]. Knowing that fact, the results shown in the maps were adapted to the current ASF zones, colors, and nomenclature.

### 2.5. Statistical Analyses

The statistical analyses were conducted with the application of logistic regression models. This type of model is a mathematical formula that can be used to report the effect of several variables (*X*_1_*, X*_2_*, …, X_n_*) on the dichotomous variable Y, which has one of two possible rates (in this case: positive or negative):$$P\left(Y=1|x_{1},x_{2},\ldots ,\;x_{n}\right)=\frac{e^{\left(\beta_{0}+\sum_{i=1}^{n}\beta_{i}x_{i}\right)}}{1+e^{\left(\beta_{0}+\sum_{i=1}^{n}\beta_{i}x_{i}\right)}}$$
where: *β_i_*—regression coefficient for *i* = 0,1,…,*n*, *x_i_*—independent variables (measurable or qualitative) for *i* = 1,2,…,*n*.

To obtain the ratings of the coefficients, the maximum likelihood method was used. The significance of the independent variables was estimated using the Wald test. The fit of the model to the data was also determined in advance using the likelihood ratio (LR statistics). Odds ratios (ORs) were established with 95% confidence intervals.

For the logistic regression model, the described relationships were statistically demonstrated at the adopted significance level of α = 0.05.

Zones II and III differed significantly from the remaining reference area—zones I and 0. Due to the low number of results and problems related to achieving model convergence, zones I and 0 were generalized to the common area “I + 0”. All statistical analyses were conducted with the use of TIBCO Software Inc. (Palo Alto, CA, USA)(2017) Statistica (data analysis software system), version 13. The yearly distributions of ASF-positive wild boars were prepared in Excel 2016 (Microsoft, Redmond, WA, USA). The geographical distribution of ASF-positive wild boar was made using ArcGIS 10.4.1 (ESRI).

## 3. Results

In the analyzed period (2014–2020), the total number of wild boars was 340,775 animals. In the study, a total of 15,639 ASF-positive wild boars and 325,136 ASF-negative wild boars were analyzed. Every year, the total number of analyzed wild boars increased compared to the year before. Most of the studied samples were ASF-negative (Figure 2a), however, the number of ASF-positive wild boars increased every year. The highest number of ASF-positive results was noted in molecular/virus analysis (qPCR-positive only). The serologically positive animals took second place (ELISA/IPT positive; qPCR negative) and that group was also increasing each year from 2017. All obtained results (quantitatively and percentage) divided into every type of result are gathered in Figure 2. All ASF molecular/virus-positive samples belonged to ASFV genotype II.

Detailed division of the ASF-positive wild boars analyzed by both type of tests (molecular/virus and serological) are presented in Figure 3. In that division, most of the ASF-positive wild boars were only serologically positive and the total number of that type of animals increased year-to-year since 2017 (Figure 3). The data indicate the increase in percentage of only serologically ASF-positive wild boars in relation to other types of ASF-positive wild boars in almost every year of the ASF epidemic in Poland in the analyzed period (Figure 3c).

In the first two years (2014–2015), ASF was limited to one Polish province: Podlaskie (Figure 4a,b). In 2016, the disease spread to two neighboring provinces: Lubelskie and Mazowieckie (Figure 4c). In 2017, the first ASF-positive wild boar was notified in Warmińsko-Mazurskie province, near the Russian border. In addition, the disease was introduced to the Warsaw region (Mazowieckie province) far away from previous ASF outbreaks (Figure 4d). The next year brought an increased number of new ASF outbreaks in wild boars. Moreover, an ASF outbreak in domestic pigs occurred in Podkarapckie province (previous ASF-free area; Figure 4e). In 2019, ASF outbreaks in wild boars were also noted in Podkarpackie province, and the virus was introduced to three more provinces (Lubuskie, Dolnośląskie, Wielkopolskie), far away from the previous outbreaks (Figure 4f). In 2020, the first ASF-positive wild boars were detected in Pomorskie and Zachodniopomorskie provinces (Figure 4g). The virus reached the Polish–German border (western border) seven years after the initial introduction of ASF to Poland from the Polish–Belarus border (eastern border).

In 2019 and 2020, an increase in the number of serologically positive results in the eastern part of Poland (Lubelskie and Podlaskie provinces) was observed, and in Warsaw cluster in 2020, in places where molecular/virus-positive results were mainly noted a year before (Figure 4f,g).

### 3.1. Logistic Regression Models for Wild Boars Analyzed in Both Tests (Molecular/Virus and Serological)

In the first logistic regression model, the dependent variable was the final serological result (positive), while the molecular/virus result (positive) was treated as an independent variable, along with the other analyzed variables: year, ASF zone, wild boar status. Only those wild boars that were tested both molecularly and serologically (224,117) were included in this analysis (Table 1).

The model showed a significant effect on the seroprevalence level of all used categories of variables: molecular result, animal status, ASF zone, and year (*p* < 0.0001).

The model indicates that the seroprevalence among the molecular-positive wild boars was significantly higher. The chance of a positive serological result in the molecular/virus-positive wild boars was more than 11 times higher (Table 1).

With the exception of 2015 (*p* = 0.93), all of the following analyzed years showed a significant difference compared to the reference year 2014. The chance of a positive serological result in 2016 was almost 4.5 times higher, in 2017 almost 5.5 times higher, in 2018 over 6.5 times higher, in 2019 5 times higher, and in 2020 it was over 5 times higher than in the reference 2014 (Table 1).

The chance of a positive serological result in zone II was over 15 times higher, while in zone III it was over 50 times higher than in the reference zone “I + 0” (Table 1).

The model showed that the seroprevalence among hunted wild boars was significantly lower than among the reference found-dead. The chance of a seropositive result in the hunted wild boars was almost 2 times lower than in the found-dead wild boars. For road-killed wild boars, no statistically significant difference could be found in relation to the found-dead animals (Table 1).

### 3.2. Logistic Regression Models for All Analyzed Wild Boars (Molecular/Virus and/or Serological)

In the second analysis, the modeled variable was the final molecular/virus result (positive), and the serological result (positive) was placed among the other variables—including the value “lack of serological analysis” when there was no serological test performed (lack of serum/blood sample from the analyzed wild boar), because all of the wild boars were already included in this analysis. It also made it possible to analyze all zones (0, I, II, III) separately, without additional generalization (Table 2) as in Section 3.

The model showed a significant influence of the molecular result on all used categories of variables: serological result, animal status, zone, and year (*p* < 0.0001).

The model indicates that the molecular/virus-positive results were significantly more frequent in serologically positive and serologically untested wild boars than in the serologically negative animals. The chance of a positive molecular/virus result in a serologically positive wild boar was more than 13 times higher than in a serologically negative wild boar, while the chance of a positive molecular result in a wild boar not serologically tested was over 3.5 times higher than in a serologically negative wild boar (Table 2).

The model showed that the percentage of molecularly/virus-positive results, both in the hunted and road-killed wild boars, was significantly lower than in the reference found-dead group. The chance of a positive molecular/virus result in a hunted wild boar was over 150 times lower than in the found-dead animals, while the chance of a positive molecular result in road-killed wild boar was over 45 times lower than in the reference group (Table 2).

Zones I, II, and III differed significantly from zone 0, in which the positive molecular/virus results were of course the lowest (ASF-free area). The chance of a positive molecular/virus result in zone I was over 13 times higher, in zone II it was almost 130 times higher, and in zone III it was almost 175 times higher than in zone 0 (Table 2).

With the exception of 2016 (*p* = 0.40), all other years showed a significant difference compared to the reference year 2014. The chance of a positive molecular/virus result in 2015 was 1.7 times lower, in 2017 it was almost 2.5 times higher, in 2018 it was over 2.5 times higher, in 2019 it was 1.8 times higher, and in 2020 it was almost twice as high as in the reference 2014 (Table 2).

The analysis of the percentage of serologically positive wild boars in relation to the animals analyzed with both types of tests is presented in Figure 5.

## 4. Discussion

An ASF epidemic started in Poland in early 2014, when the first ASF-affected wild boar was found near the Belarusian border [26]. ASFV has been present in Poland since then. In 2014, the disease was also notified in other Baltic States (Lithuania, Latvia, and Estonia) where the number of ASF-positive wild boars at the beginning of the epidemic was low, but the number increased over the next years [27], similarly to Poland.

Surveillance is very important in monitoring the progress of ASF in the wild boar population. There are two main groups of action focused on the monitoring: passive and active surveillance. The first is connected with the virologic (molecular) analysis of the collected carcasses of dead animals from the environment (i.e., forests). The second is focused on analysis of the hunted wild boars [27,28,29]. 

The material from the animals found dead rarely includes blood samples, which excludes serological analysis. In the analyzed period of the ASF epidemic in Poland, most ASF-positive samples were molecular/virus-positive. The previous analysis indicated in Poland confirms that most positive wild boars were from passive surveillance [26,28,29,30]. Similarly, in Lithuania, the number of ASF-positive wild boar carcasses increased from 20.2% in 2014 to 79.7% in 2017 [31]. In successive years, the number of wild boar outbreaks decreased, however, the total number of ASF-positive animals was higher via passive surveillance than active [12]. The Estonian course of ASF processed similarly. In the first three years of the epidemic (2014–2016), the number of ASF-positive wild boars was increasing and most of them were notified via passive surveillance [32], as in Poland. After 2016, the number of new wild boar outbreaks in Estonia decreased and was mainly connected with new areas of ASF [33], as is observed in our analysis in the case of Poland. In 2019 and 2020 there were no molecular/virus ASF-positive wild boars in Estonia, as mentioned before [10]. In 2014–2017 in Latvia, most ASF-positive results were noted in carcasses [34], however, the number of molecular/virus-positive wild boars has started to decrease since the 2017/2018 hunting season [35]. In all Baltic States, like in Poland, most ASF-positive animals were confirmed via passive surveillance, at least in the first years of the epidemic.

Active surveillance, strongly connected with serological monitoring (but not only) of hunted wild boars, brought interesting results in the Baltic States and in Poland. As mentioned before, most ASF-positive animals in the Baltic States were virus-positive in the first years of the epidemic [31,32,34], however, the situation has changed over the years. In Poland (Figure 3), an increase of serologically positive animals in active surveillance in the analyzed period was observed (2014–2020). Since May 2019 in Lithuania, the number of virus-positive wild boars has started decreasing and in the same period the number of seropositive animals has started to increase [12]. Estonian researchers have noted a decreasing number of molecularly-positive wild boars in the regions where ASFV has been present for a longer time. In the same time period, in the same area, the number of serologically positive animals has started increasing [33]. The same situation was observed in Poland (Figure 4). The researchers indicate that the observed course of the disease could be the beginning of an ASF endemicity in Estonia [33]. From February 2019 to October 2020 in Estonia, all ASF-positive wild boars were noted only in the active surveillance group, analyzed by serological methods [10]. In successive years, the growth of seroprevalence simultaneously with a decrease of virus-positive animals was also identified in Latvia. The researchers found that the prevalence of anti-ASF antibodies was higher in older wild boars than in younger ones (below 1 year). At the same time, the presence of ASFV DNA was higher in the younger group. The Latvian data indicate that the course of ASF leads to the death of most of the animals and the survivors cumulate in time [35].

The report on the data from Poland described the potential role of younglings (below 1 year) in the serologically positive wild boar group. In the period of 2017–2018 in Poland, 292 animals with anti-ASF antibodies were confirmed. In this group, 126 of them were from young wild boars. The researchers explain this phenomenon as the potential presence of natural survivors and potential vertical transmission of antibodies through the colostrum [36].

In the Baltic States, where the ASF epidemic started in the same year as in Poland, most ASF-positive results are currently noted in serological monitoring. In Poland, an increase of serologically positive wild boars is observed, however, there is still a large number of virus-positive animals. This discrepancy can be explained by the size of the countries. In the Baltic States, almost the whole area of the countries is covered by ASF zones, while in Poland about half of the territory of the country still remains ASF-free, so the disease was able to spread to new areas. In Figure 5, the progress of seroprevalence in wild boars is presented. Knowing the situation in the Baltic States, the graph should be expected to be straighter, however, there are two slight refractions in 2018 and 2019 which were connected with the long distance of the new ASF introduction. The decrease in 2018 was connected with two new introductions: to Warsaw cluster in late 2017 and to the northern part of Poland (near the Russian border) [30]. The decrease in 2019 was the result of large-scale introduction of the virus to Lubuskie province [37]. The new ASF introduction brought a large number of new virus-positive wild boars.

The increase of seropositive wild boars in relation to virus-positive animals, as was suggested by Estonian researchers [33], might lead to the endemicity of ASF in the analyzed region. Endemic diseases are often mistaken for epidemics. Epidemic refers to an outbreak of a disease which spreads through one or more populations. In the case of an endemic disease, the disease is constantly present in a group of animals (in our case: in wild boars), or in geographic area. Multiple sporadic cases of the disease that occur in time and space might be referred as endemic disease [38].

ASF originates from Africa and is still present on that continent. The phylogenetic analysis of all ASFV genotypes indicates that the last common ancestor of all of them was in the 18th century [39]. The virus has had enough time to adapt to different hosts. Currently, ASF is endemic in most of the sub-Saharan countries where wild hosts (*Suidae*) and soft tick vectors from the *Ornithodoros* genus play an important role as the biological reservoirs of ASFV [6]. In the case of Europe, ASF (since 1978) is still present in Sardinia (Italy), and the disease is considered to be endemic there [6,40,41]. For Sardinia, ASF becoming endemic was potentially connected with areas of certain socio-economic factors (low income/salary), the high density of wild boars, and the traditional farming practice (free-range herds: brado pigs) without veterinary controls [40]. The latest study from Italy indicates the role of ASF survivors in the endemicity of ASF in Sardinia. In the years 2017–2020, 4484 pigs from illegal farms were culled. In this group, 36.5% of them had antibodies against ASFV, while only 53 of these animals were virus-positive [41].

Intensive carcass removal is the most significantly import part of disease control, especially in the endemic phase. Transmission of the virus through infected carcasses accounts for about 53–66% of cases [42]. Susceptible animals have contact with contaminated carcasses at every stage of decay, where the virus is stable for a few months [15,20]. It is estimated that contaminated carcasses, along with direct virus transmission between infected and susceptible animals, increase the ability of ASF to persist in wild boars even at low population densities [1]. The effectiveness of this measure has been confirmed by the Czech Republic and Belgium [1,43]. In implementation of effective control strategies, it should be remembered that ASFV can also spread through vectors, contact with fomites, contaminated feed, water, and soil [42].

ASF is still spreading across the territory of Poland, however, it has still remained active in the original area for years. The number of seropositive animals has been growing in successive years, like in the Baltic States [10,12,35]. The percentage of seropositive animals is still at a low level, however, the percentage is connected with new ASF outbreaks in wild-boars in new areas. The received data might indicate the potential beginning and progress of the endemic, however, the current ASF status in neighboring countries (Russia, Belarus, Ukraine) is not clear, and the genetic data indicate continuous reintroductions from neighboring countries to Poland [37]. A new virus transmission might interfere with the local wild boar population, creating a new epidemic across the borders.

## 5. Conclusions

The seven years of ASF in Poland has brought an increase of new ASF wild boar outbreaks in successive years. At the beginning of the epidemic, the disease was limited to a small area, however, it spread to new provinces with time. From the beginning and until today, most ASF-positive wild boars were noted in the carcass groups (passive surveillance), however, the number of serologically positive animals is increasing. In 2022, ASF zones are present in every Polish province [13]. Despite the fact that Poland still confirms ASF outbreaks near its eastern border, the “old” ASF area seems to contain most of the ASF serologically positive animals, which may indicate the beginning of ASF endemicity in Poland.

Continuous observation of ASF outbreaks in wild boar population indicate that in the area where the ASF is present for a longer time indicate that the disease start to being endemic in Poland.

## Figures and Tables

**Figure 1 animals-12-01170-f001:**
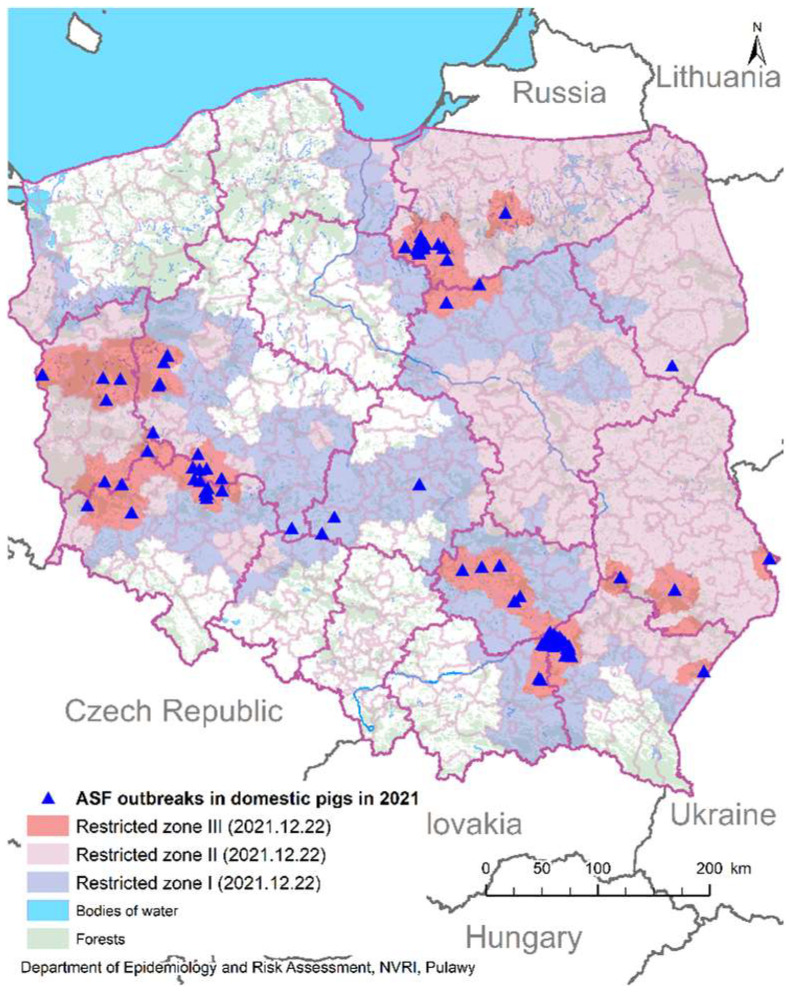
ASF outbreaks in domestic pigs in Poland in 2021.

**Figure 2 animals-12-01170-f002:**
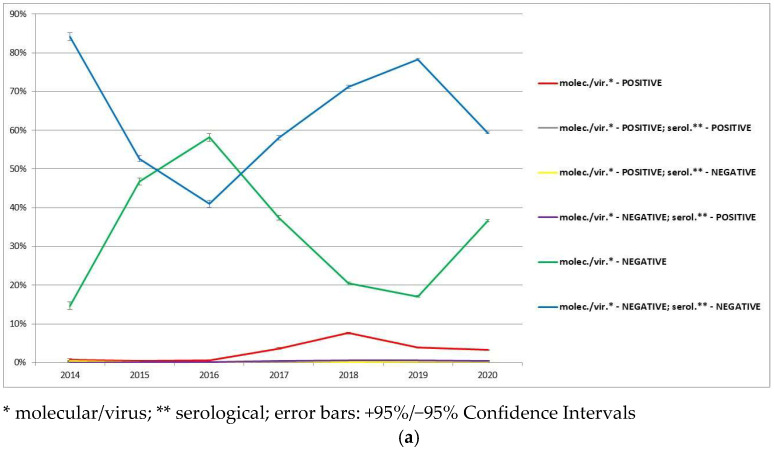
Wild boars tested for the presence of ASFV and/or antibodies against ASFV in Poland in the years 2014–2020: (**a**) percentage division of all analyzed wild boars; (**b**) percentage division of analyzed wild boars without ASF-negative animals.

**Figure 3 animals-12-01170-f003:**
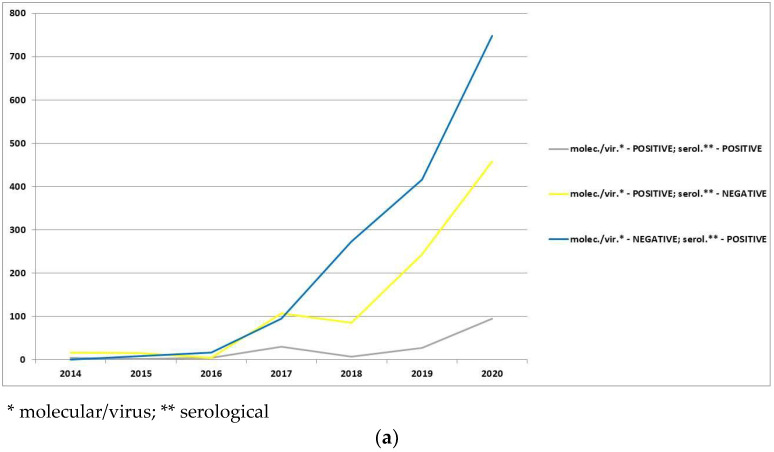
ASF-positive wild boars analyzed by both type of tests (molecular/virus and serological): (**a**) quantitative division of ASF-positive wild boars; (**b**) percentage of ASF-positive wild boars in relation to all analyzed wild boars; (**c**) percentage of ASF-positive wild boars in relation to wild boars analyzed with both type of tests.

**Figure 4 animals-12-01170-f004:**
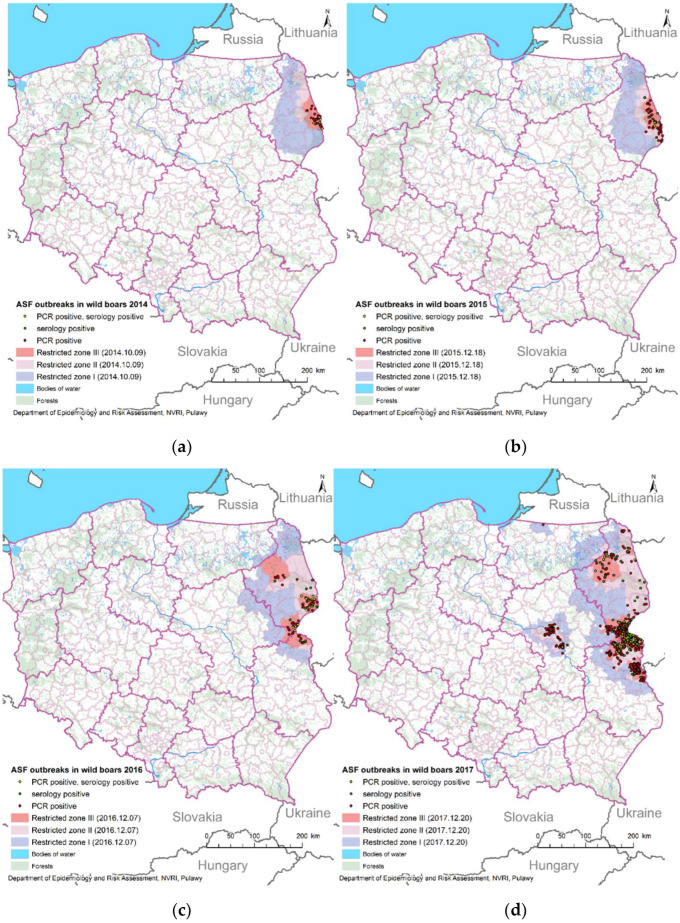
ASF-positive wild boars in Poland in: (**a**) 2014; (**b**) 2015; (**c**) 2016; (**d**) 2017; (**e**) 2018; (**f**) 2019; (**g**) 2020.

**Figure 5 animals-12-01170-f005:**
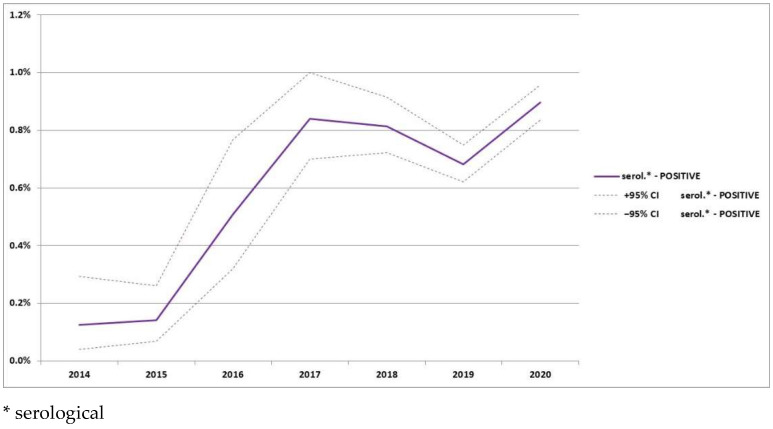
Percentage of ASF-positive wild boars in relation to wild boars analyzed with both types of tests in years 2014–2020.

**Table 1 animals-12-01170-t001:** Results of logistic regression models—impact of the type of result, animal status, ASF zone, and the year for the final serologically positive result (reference type of result: serologically positive; reference animal status: found-dead; reference ASF zone: zone 0 + I; reference year: 2014).

Significance Assessment of Model (*p* Value of LR ^1^ Test)	Independent Variable	Coefficient (βi)	Std. ^2^ Error	*p* Value (Wald)	Odds Ratio	Confidence OR ^3^—95%	Confidence OR ^3^ + 95%
<0.0001	Absolute term (β0)	−8.765	0.520	<0.001	<0.001	<0.001	<0.001
	Molecular/ virus-positive	2.416	0.109	<0.001	11.201	9.037	13.884
	Hunted	−0.590	0.154	<0.001	0.555	0.409	0.751
	Road-killed	−0.649	0.531	0.222	0.523	0.183	1.491
	Zone II	2.733	0.187	<0.001	15.371	10.631	22.225
	Zone III	3.966	0.187	<0.001	52.765	36.477	76.327
	2015	−0.053	0.580	0.927	0.985	0.302	2.979
	2016	1.460	0.514	0.005	4.305	1.563	11.858
	2017	1.672	0.475	<0.001	5.321	2.083	13.592
	2018	1.900	0.469	<0.001	6.687	2.651	16.868
	2019	1.612	0.468	<0.001	5.012	1.992	12.612
	2020	1.626	0.466	<0.001	5.084	2.025	12.761

^1^ LR—logistic regression; ^2^ Std.—standard; ^3^ OR—odds ratio.

**Table 2 animals-12-01170-t002:** Results of logistic regression models—impact of the type of result, animal status, ASF zone, and the year for the final molecular/virus-positive result (reference type of result: molecular/virus-positive; reference animal status: found-dead; reference ASF zone: 0; reference year: 2014).

Significance Assessment of Model (*p* Value of LR ^1^ Test)	Independent Variable	Coefficient (βi)	Std. ^2^ Error	*p* Value (Wald)	Odds Ratio	Confidence OR ^3^—95%	Confidence OR ^3^ + 95%
<0.0001	Absolute term (β0)	−6.074	0.200	<0.001	0.002	0.002	0.003
	Serologically positive	2.580	0.110	<0.001	13.191	10.603	16.411
	Lack of serological analysis	1.267	0.058	<0.001	3.551	3.163	3.985
	Hunted	−5.039	0.058	<0.001	0.007	0.006	0.007
	Road-killed	−3.812	0.072	<0.001	0.022	0.019	0.026
	Zone I	2.589	0.107	<0.001	13.311	10.777	16.440
	Zone II	4.848	0.096	<0.001	127.463	105.455	154.064
	Zone III	5.162	0.097	<0.001	174.460	143.951	211.436
	2015	−0.504	0.221	0.023	0.604	0.390	0.936
	2016	−0.193	0.227	0.395	0.825	0.527	1.292
	2017	0.883	0.179	<0.001	2.419	1.698	3.447
	2018	0.952	0.173	<0.001	2.591	1.841	3.645
	2019	0.580	0.172	<0.001	1.786	1.270	2.511
	2020	0.67543	0.172	<0.001	1.965	1.400	2.760

^1^ LR—logistic regression; ^2^ Std.—standard; ^3^ OR—odds ratio.

## Data Availability

Not applicable. Data are contained within this article.

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
