# Peer review of "African Swine Fever (ASF) Trend Analysis in Wild Boar in Poland (2014–2020)"

_animals, 2022, doi:10.3390/ani12091170_

Round 1

Reviewer 1 Report

Dear authors

I find this manuscript ("African swine fever (ASF) trend analysis in wild boar in Poland (2014–2020)")  very interesting and easy to read. However, some small flaws can be improved. In general, I find it too long for a research paper. I recommend some changes:

  1. This manuscript needs an objective in the last paragraph of Introduction. If the final conclusion is "...the “old” ASF area seems to contain most of the ASF serologically positive animals, which may indicate the beginning of ASF endemicity in Poland." (lines 616-618), the last sentence  in introduction should indicate the intention of the authors to analyze the real state of ASF in Poland at this moment (2020) and the evolution in time since the first observation.
  2. Materials and methods should be divided in parts with titles to to facilitate the understanding. I recommend for instanceLines 152-160: 2.1. Sampling methods
    Lines 161-176: 2.2. PCR technique
    Lines 177-188: 2.3. Serological analyses
    Lines 189-225: 2.4. Survillance
    Lines 226-249: 2.5. Statistical analyses
  3. Results
    Figure 1. I find it too bigand also difficult to understand. I think that a and b parts should be eliminated, as the absolute numbers are difficult to compare and maintain only c and d, as proportions (percentages in this case) are easier to compare. I think also that it shoulb be better to use other colors that contrast more (something like intense colors as red, blue, yellow and green)
    Figure 2. Legend is not complete. If the values are proportions, what do the error bars represent? This must be indicated in legend.
    Figure 3 is to large. I think it would be better to reduce each image, try to put them two by two and fit them in a single page, to see all in one view and compare the temporal evolution as a whole.
    Table 1. Reduce decimals to 3, including this 0 that should be <0.001
    Lines 358-362 must be included in Materials and methods

  4. Including this part, 3.3. Analysis of the percentages of combinations of results (molecular/virus and serological) among wild boars with respect to year (chi-square test), has no sense to me. you have analyzed a multivariate logistic model and then you introduce a collection of univariate analyses? For what? you have obtained these information in 3.4. In any case, I recommend to eliminate completely 3.3 and reduce 3.4 to figure 4, indicating some information with text and indications in figure 4. I think this manuscript is too long. So, table 3 and 4 shoulb be eliminated also, including the main information in the text and figure 4. I also think that this information could be included after Table 2, as it is also a rotation of the reference category in the same analysis. This information is included in the analyses realized in table 2, including rotation in reference year, without a new and univariate logistic regresion.

Author Response

Dear Reviewer thank you for yours valuable comments. Your suggestions has enrich our work and made it more clear for the reader.

Responding to your comments:

“1. This manuscript needs an objective in the last paragraph of Introduction. If the final conclusion is "...the “old” ASF area seems to contain most of the ASF serologically positive animals, which may indicate the beginning of ASF endemicity in Poland." (lines 616-618), the last sentence  in introduction should indicate the intention of the authors to analyze the real state of ASF in Poland at this moment (2020) and the evolution in time since the first observation.”

  • The suggestion of the Reviewer is true and important for the whole manuscript. We have add the sentence at the end of the Introduction:

“When the disease is present for a long time in the same area, the animals may adapt to virus pressure in the environment. The changes observed in the serological results during analyzed period might prove potential endemicity of the ASF in Poland.”

“2. Materials and methods should be divided in parts with titles to to facilitate the understanding. I recommend for instance

Lines 152-160: 2.1. Sampling methods

Lines 161-176: 2.2. PCR technique

Lines 177-188: 2.3. Serological analyses

Lines 189-225: 2.4. Surveillance

Lines 226-249: 2.5. Statistical analyses”

  • As was suggested the Material and methods section was adapted to the Reviewer guidelines.

“3. Results

Figure 1. I find it too big and also difficult to understand. I think that a and b parts should be eliminated, as the absolute numbers are difficult to compare and maintain only c and d, as proportions (percentages in this case) are easier to compare. I think also that it should be better to use other colors that contrast more (something like intense colors as red, blue, yellow and green)”

  • Changed as Reviewer suggested.

“Figure 2. Legend is not complete. If the values are proportions, what do the error bars represent? This must be indicated in legend.”

  • As in case Figure 1 (new No 2) colours were adapted as Reviewer suggested. In Legend was added information:

“error bars: +95%/-95% Confidence Intervals”

“Figure 3 is to large. I think it would be better to reduce each image, try to put them two by two and fit them in a single page, to see all in one view and compare the temporal evolution as a whole.”

  • The images were reduced as suggested.

“Table 1. Reduce decimals to 3, including this 0 that should be <0.001”

  • The Table was changed as suggested. The same was done with Table 2.

“Lines 358-362 must be included in Materials and methods”

  • The mentioned section was moved to the paragraph “2.5. Statistical analyses”.

“4. Including this part, 3.3. Analysis of the percentages of combinations of results (molecular/virus and serological) among wild boars with respect to year (chi-square test), has no sense to me. you have analyzed a multivariate logistic model and then you introduce a collection of univariate analyses? For what? you have obtained these information in 3.4. In any case, I recommend to eliminate completely 3.3 and reduce 3.4 to figure 4, indicating some information with text and indications in figure 4.”

  • The paragraph 3.3 was removed + information about that analysis from Materials and methods section was removed.

“I think this manuscript is too long. So, table 3 and 4 should be eliminated also, including the main information in the text and figure 4. I also think that this information could be included after Table 2, as it is also a rotation of the reference category in the same analysis. This information is included in the analyses realized in table 2, including rotation in reference year, without a new and univariate logistic regresion.”

  • Paragraph 3.4 with the Tables 3-4 was removed and the information was reduced to one sentence at the end of paragraph 3.2.

Sincerely,

Authors

Reviewer 2 Report

Comments to the Authors of manuscript number: animals-1696267 entitled “African swine fever (ASF) trend analysis in wild boar in Poland (2014–2020)”.

The authors have presented the analysis of samples from wild boars related to the ASF monitoring in the years 2014-2020. All the analyses were performed in a biosafety 3 laboratory for animals and research which must be certified for use before initial operation. It is very good description.

  1. L 47-49 this sentence should be rephrased and cannot be end by etc.
  2. L 74 – there should be given a small map, not description, because readers do not know how Poland looks. And the total number of provinces should be given, because I do not know if it is large number or not in relation to total number
  3. Introduction should be about the ASF reports, and carcass removal should be shift to discussion

Author Response

Dear Reviewer thank you for your precious comments. Your opinion is very important for us and will enrich our article.

Responding to your comments:

“L 47-49 this sentence should be rephrased and cannot be end by etc.”

  • The sentence was rebuild as suggested:

“Infected animals show a wide range of symptoms depending on virus isolate, the route and dose of infection and the immunological status of the animal.”

“L 74 – there should be given a small map, not description, because readers do not know how Poland looks. And the total number of provinces should be given, because I do not know if it is large number or not in relation to total number”

  • As Reviewer has suggested the sentence was rebuild and was added a new Figure 1:

“In Poland in 2021, a large number of outbreaks in domestic pigs were reported in eleven provinces out of sixteen (Figure 1).”

“Introduction should be about the ASF reports, and carcass removal should be shift to discussion”

  • As Reviewer suggested “carcass removal” part was moved to the Discussion section.

Sincerely,

Authors

Round 2

Reviewer 1 Report

Dear authors,

I think this volume is more adequate for a research paper and now it is also easier to understand.

Best regards,